evolution, palaeontology, taxonomy and systematics

phylogeny, morphology, palaeontology, fossils, Bayesian inference, tip-dating

**Author for correspondence:**
Nicolás Mongiardino Koch
e-mail: nicolas.mongiardinokoch@yale.edu

# Fossils improve phylogenetic analyses of morphological characters

Nicolás Mongiardino Koch[1], Russell J. Garwood[2,3] and Luke A. Parry[4]

[1]Department of Earth and Planetary Sciences, Yale University, New Haven, CT, USA
[2]Department of Earth and Environmental Sciences, University of Manchester, Manchester, UK
[3]Earth Sciences Department, Natural History Museum, London, UK
[4]Department of Earth Sciences, University of Oxford, Oxford, UK

NMK, 0000-0001-6317-5869; RJG, 0000-0002-2803-9471; LAP, 0000-0002-3910-0346

Fossils provide our only direct window into evolutionary events in the distant past. Incorporating them into phylogenetic hypotheses of living clades can help time-calibrate divergences, as well as elucidate macro-evolutionary dynamics. However, the effect fossils have on phylogenetic reconstruction from morphology remains controversial. The consequences of explicitly incorporating the stratigraphic ages of fossils using tip-dated inference are also unclear. Here, we use simulations to evaluate the performance of inference methods across different levels of fossil sampling and missing data. Our results show that fossil taxa improve phylogenetic analysis of morphological datasets, even when highly fragmentary. Irrespective of inference method, fossils improve the accuracy of phylogenies and increase the number of resolved nodes. They also induce the collapse of ancient and highly uncertain relationships that tend to be incorrectly resolved when sampling only extant taxa. Furthermore, tip-dated analyses under the fossilized birth–death process outperform undated methods of inference, demonstrating that the stratigraphic ages of fossils contain vital phylogenetic information. Fossils help to extract true phylogenetic signals from morphology, an effect that is mediated by both their distinctive morphology and their temporal information, and their incorporation in total-evidence phylogenetics is necessary to faithfully reconstruct evolutionary history.

## 1. Introduction

Phylogenies underpin our ability to make sense of life on Earth in the context of its shared evolutionary history. In the absence of phylogenetic hypotheses, we would be unable to explain the myriad of biological phenomena that arise as the result of common ancestry, such as shared morphological features among seemingly disparate taxa. Many analyses that seek to reconstruct past evolutionary events do so using data from only living organisms. However, this might not be enough to faithfully recover events that occurred in the distant past, as extant-only trees often lack the information necessary to distinguish between alternative scenarios [1] and can even favour incorrect results [2]. The incorporation of fossils into comparative analyses has been shown to have a positive effect on the inference of modes of macroevolution, ancestral character states and patterns of speciation and extinction [1–4]. Placing fossils in a phylogenetic framework is therefore necessary not only to understand their affinities but also to obtain an accurate picture of evolutionary history.

The effect that adding fossils has on phylogenetic inference remains equivocal, however. Although fossils were originally dismissed as being too fragmentary to modify inferred relationships among living species [5], a number of empirical studies have demonstrated a common pattern of increased congruence between morphological and molecular phylogenies when palaeontological data are introduced (e.g. [6–9]). A recent analysis of multiple empirical datasets showed that adding fossil taxa to morphological matrices reshapes phylogenies in a manner that is entirely distinct from increasing the sampling of extant taxa [10], a result

largely attributable to the possession of distinctive character combinations in fossil taxa. However, given that the true tree of life is unknown, this study was unable to determine if these topological changes resulted in phylogenies that are more accurate.

Distinct combinations of morphological characters are not the only potential source of phylogenetic information that fossils can provide. The stratigraphic sequence of taxa in the rock record has also been proposed as a source of data with which to infer phylogenies, as taxon first appearances should reflect their phylogenetic position if the fossil record were perfect [11,12]. Although the use of fossil ages in the process of tree inference was first formalized in a controversial parsimony framework known as stratocladistics [13], it has since become popular again through the development of Bayesian tip-dated methods [14]. In this new framework, tree topology and divergence times are simultaneously estimated using the ages of fossil terminals to calibrate a morphological and/or molecular clock. This approach has since developed to incorporate mechanistic models that include parameters for the rates of speciation, extinction and taxon sampling in an attempt to model the macroevolutionary process that generated the data, i.e. the fossilized birth–death process (FBD) [15,16]. Although the estimation of evolutionary timescales was the motivation for the development of these methods, they have also been found to strongly modify inferred topologies [17], thus reshaping our understanding of evolutionary relationships. The use of temporal information from the fossil record as data in phylogenetic inference has been criticised however, with concerns ranging from the incompleteness of the geological record [18] to the non-clocklike evolution of morphology [19].

Inferring the phylogenetic position of fossils can only be achieved using morphological datasets. Methods for inferring phylogenies from discrete morphological characters have been increasingly scrutinised in recent years, establishing Bayesian inference (BI) as a valid alternative to parsimony approaches [20–24]. However, much of the early research on this topic has been criticized for simulating data under models of morphological evolution that are similar to those used in BI, thus potentially biasing results [25]. In a similar vein, another study recently found tip-dated trees of extinct clades to be superior to undated ones [26], a conclusion drawn from topologies simulated under the same birth-death processes used for tip-dated inference. Empirical analyses suggest fossils may overcome many limitations of morphological data for inferring the tree of life, but none of the above simulation studies explored the topological effects of sampling extinct lineages—or their associated missing data.

In this study, we employ a simulation approach [24] to obtain character datasets and associated phylogenies that does not rely on any model later employed in the process of phylogeny reconstruction, and thus should not unduly favour any inference method. This also introduces a level of model misspecification that is common to the analysis of all empirical morphological datasets. We fine tune our simulations to produce mixtures of living and extinct taxa, and ensure our trees and characters are empirically realistic through comparison with published morphological datasets, following best practices for palaeobiological simulation [27]. Furthermore, we also mimic the presence of an ancient and rapid radiation [28]: a tree shape comparable to that of the early radiation of placental mammals [29] or the origin of most modern animal phyla during the Cambrian explosion [30], as it is often suggested that fossils might be especially beneficial under this scenario [31]. With these datasets we are able to explore the impact of palaeontological data on the accuracy of inferred phylogenies, to our knowledge for the first time. We do this through sampling different proportions of fossil taxa across a range of conditions of missing data, and investigate the relative behaviour of tip-dated inference under the FBD relative to traditional undated approaches, i.e. maximum parsimony and undated BI.

## 2. Material and methods

We used TREvoSim v. 2.0.0 [24,32], newly released with this paper (binaries and code available at [33], code archived at [32]), to simultaneously simulate 250 phylogenies and associated character matrices (a graphical summary of the procedure is shown in the electronic supplementary material, figure S1). TREvoSim is an individual-based simulation that incorporates natural selection and implements its own species definition. It does not rely on either Markov models for the generation of character datasets, nor birth–death processes for the simulation of topologies, both of which are emergent properties of the simulation, and thus should be unbiased towards alternative inference approaches. Simulations were composed of 500 binary characters, and were run until they comprised 999 terminals (the maximum allowed), using settings specified in file S1 [34] (further details are reported in the detailed methods, electronic supplementary material). The resulting phylogenies contained a mean of 150 extant terminals at termination and exhibited a range of tree symmetries (electronic supplementary material, figure S2), as estimated using Colles' index [35]. We removed fossils at random from these until only 300 terminals remained, a step that emulated the reality that a significant amount of extinct biodiversity is not captured in the fossil record. These reduced datasets hence constitute true and theoretically knowable evolutionary histories. Other simulation settings were chosen to emulate realistic properties of morphological datasets, including rates of evolution, distribution of branch lengths, and levels of treeness (i.e. the fraction of total tree length that is on internal branches [36]), based on a comparison with 12 empirical datasets [34] (see also the electronic supplementary material, table S1 and figures S3 and S4). As previously outlined, our chosen settings generate topologies with a series of deep and short internodes: tree shapes comparable to that of clades that underwent an ancient and rapid radiation [28]. Simulated datasets are available in file S2 [34].

Through subsampling, we built a total of 11 250 morphological matrices from these simulations, varying the levels of missing data and proportions of fossil terminals (see the electronic supplementary material, figure S5, for a summary of this procedure). All resulting matrices were composed of 100 terminals and 300 parsimony-informative characters. Terminals were selected at random from among the available fossil and extant tips, in such a way as to obtain matrices with five levels of fossil sampling: 0, 10, 25, 50 and 100%. Furthermore, datasets were analysed without any amount of missing data, as well as implementing low (25% for extant taxa, 37.5% for fossils) and high levels of missing data (25% and 50%, respectively). The latter condition was designed to mimic realistic levels of missing data as found across the empirical datasets surveyed (electronic supplementary material, table S1). Imputation of missing data was performed at random, resulting in some variation in the final number of characters coded per taxon. Three iterations of each condition were performed per original dataset, reducing the topological effects that are merely a consequence of character and taxon sampling. Manipulation of topologies and character matrices was performed within the R environment [37] using custom scripts [34] that make use of packages *ape* [38], *Claddis* [39] and *phytools* [40]; morphological matrices are available as file S3 [34].

All matrices were analysed using equal weighted maximum parsimony (henceforth MP), as well as both undated (BI) and tip-dated (clock) Bayesian approaches. In the latter case, we implemented either the fossilized birth–death [15] or birth–death tree priors, depending on whether fossil taxa were sampled or not, respectively. Parameters for tip-dated analyses, such as prior distributions for the tree height and tip ages, were informed using data mined from the Fossil Calibration Database [41] (http://fossilcalibrations.org) and the Paleobiology Database (https://paleobiodb.org/), respectively. Phylogenetic inference was performed using TNT 1.6 [42] and MrBayes 3.2 [43].

We summarize these analyses using standard consensus methods (i.e. strict consensus for MP, majority-rule consensus for probabilistic methods; all available as file S4 [34]), and compared the inferred consensus topologies to true (simulated) trees using both bipartition and quartet-based measures of precision and accuracy. We define topological precision as the number of resolved bipartitions/quartets, and topological accuracy as the proportion of these that are correct (i.e. present in the true tree). While bipartition-based metrics provide an overall summary of the patterns seen across the tree, quartet-based metrics will primarily capture topological changes occurring close to the root, as deeper nodes account for a large fraction of total quartets. Furthermore, the overall performance of alternative methods of inference under different conditions was summarized using quartet distances [23] between estimated and true trees. Quartet distances have been found to outperform measures of tree similarity based on bipartitions, especially when the topologies being compared are not fully bifurcating [23,44]. They are also less susceptible to influence from wildcard taxa and tree shape, and have been found to produce more intuitive results and be less prone to saturation, than symmetric distances based on bipartitions [23,45]. Significant differences were explored by comparing the distributions of quartet distances between tip-dated inference and the best undated method across conditions, using t-tests and implementing the Benjamini & Hochberg correction for multiple comparisons [46].

We further explored the effects of incorporating fossils on relationships among living clades. We pruned simulated trees down to the subset of extant terminals and defined three time-slices of equal duration, representing deep, mid and shallow divergences (see the electronic supplementary material, figure S6). Nodes falling into each of these categories were compared with those present in the inferred consensus topologies and classified as being resolved correctly, incorrectly, or otherwise left unresolved (i.e. forming part of a polytomy). By repeating this procedure across levels of fossil sampling, we were able to isolate the effect of fossils on the resolution of extant relationships. Finally, we also explored the different ways in which inference methods resolved the position of fossil terminals (i.e. correctly/incorrectly/unresolved) depending on their relative ages. In this case, fossils were binned into 20 time-bins spanning the total depth of simulated topologies.

All analyses were performed using R code available from the Dryad Digital Repository: https://doi.org/10.5061/dryad.4xgxd2585 [34], and relied on packages mentioned previously, as well as functions from *phangorn* [47], *Quartet* [48,49] and *TNTR* [50]. Further details on data simulation, tree inference and statistical analyses, as well as supplementary figures and tables, can be found in the electronic supplementary material.

## 3. Results

Fossil terminals increase the accuracy of phylogenetic reconstruction (i.e. the proportion of correct phylogenetic statements) across all inference methods (figure 1). With no missing data, quartet-based accuracy consistently improves with an increasing proportion of fossils. Measured through bipartitions, accuracy is generally highest when 50% of terminals are fossils, and decreases with either higher or lower proportions of extinct taxa. Missing data has no effect on quartet-based accuracy, and thus has little impact on the correct resolution of deep relationships. For bipartition-based accuracy, a noticeable impact of missing data is seen for both MP and clock, but undated BI is relatively more robust.

In the absence of missing data, topological precision increases with fossil sampling when measured using bipartitions, but decreases under quartets. These opposite patterns are the consequence of higher fossil sampling improving the overall resolution of topologies (i.e. the total number of resolved nodes), but at the same time inducing the collapse of a small number of deep nodes (which are present in a large proportion of quartets). This is further supported by results shown in the electronic supplementary material, figure S7. Missing data does not modify these general trends, but does have a strong impact on overall levels of resolution (i.e. bipartition-based precision), and strongly affects both inference under MP and the analysis of entirely extinct clades. Once again, the resolution of deep relationships is stable across levels of missing data.

Average quartet distances between inferred and true trees reveal that the general performance of probabilistic approaches (clock and BI) improves when living and fossil taxa are combined, relative to their behaviour when datasets are composed entirely of either type of terminal (figure 2). MP on the other hand remains unaffected by the proportion of fossils when no data are missing, but its performance declines with increasing proportions of incomplete terminals. Across all conditions explored, probabilistic approaches recover topologies more similar to the true tree than parsimony, a difference that widens with increasing missing data (electronic supplementary material, figure S8). Tip-dating significantly outperforms uncalibrated approaches whenever fossil terminals are sampled (except in the analysis of entirely extinct clades with high levels of missing data). For any given level of missing data, the best results (i.e. the shortest distances between inferred and true trees) are always obtained when datasets combine fossil and extant terminals, and are tip-dated under the FBD model (figure 2). However, the relative benefit of tip-dating diminishes as the proportion of missing data increases (electronic supplementary material, figure S8).

Given the apparent positive reinforcement between fossil and extant terminals, we explored the effect that fossil addition has on relationships among extant terminals. Across inference methods, fossils help recover true relationships for mid and shallow divergences (figure 3), but do not affect the proportion of correctly resolved deep nodes (including those involved in ancient and rapid radiations). For undated inference methods (MP and BI), these deep nodes are predominantly resolved incorrectly, regardless of fossil sampling. However, fossils induce the collapse of deep nodes, strongly reducing topological inaccuracy. This effect is biggest when performing tip-dated inference, and is a major reason why this approach outperforms undated methods of inference in our study. By contrast, fossil placement in tip-dated trees is less accurate than under undated BI, especially so for younger fossils (electronic supplementary material, figure S9).

## 4. Discussion

Through simulations, we demonstrate that palaeontological data (both morphological and stratigraphic) have a strong

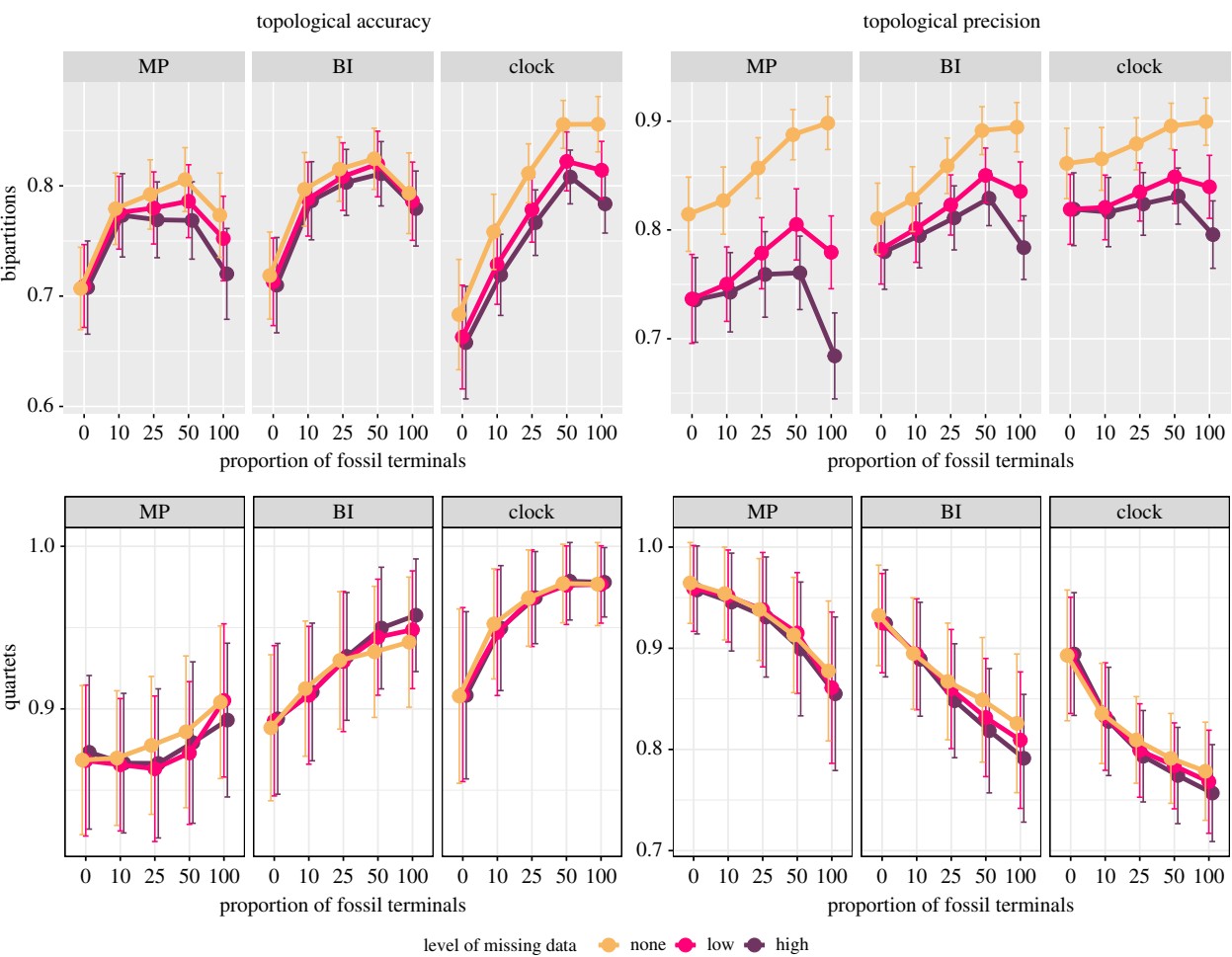

**Figure 1.** Impact of fossil sampling, missing data and method of inference on topological accuracy and precision. Precision (right) represents the proportion of resolved bipartitions/quartets, accuracy (left) the fraction of these that are correct. Values correspond to means ± 1 s.d. (Online version in colour.)

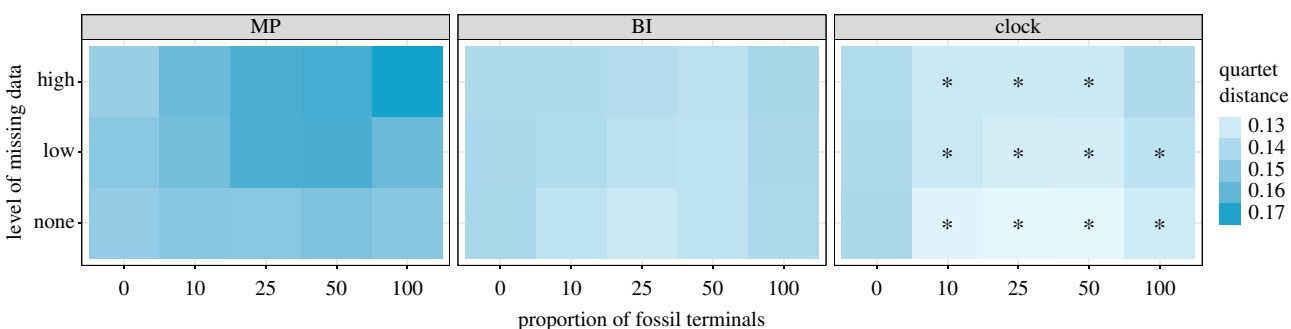

**Figure 2.** Average quartet distances between inferred and true trees across inference methods, proportions of fossil terminals and levels of missing data. For all conditions explored, clock analyses infer trees that are the most similar to the true trees (i.e. show the smallest distances), and MP the most dissimilar (i.e. show the largest distances). Asterisks mark the conditions under which the quartet distances of clock analyses are significantly lower than those of the second best method (i.e. BI). (Online version in colour.)

impact on tree inference across a wide range of realistic scenarios, which is congruent with the results of empirical studies [10]. Analyses that incorporate palaeontological data are more accurate than those based exclusively on extant taxa, regardless of inference method (figure 1). In part, this improvement is driven by fossils' power to elucidate relationships within living clades, especially among lineages separated by mid- to shallow divergences (figure 3). Hence, we might expect the increased congruence between morphological and molecular trees found for some clades [6–9,51] to reflect a general trend of consilience through improved accuracy as fossils are incorporated in phylogenetic

reconstruction. Trees that combine living and extinct taxa also show a higher proportion of resolved nodes, while at the same time leaving more deep nodes unresolved (figures 1 and 3). The phylogenetic analysis of morphological data has been previously shown to result in overprecise topologies [21,24], a phenomenon we find to be most prevalent among deep divergences. This result implies that characters evolving at rates comparable to those of empirical morphological traits fail to retain phylogenetic signal for ancient and rapid divergences (electronic supplementary material, figures S3 and S10). With increasing fossil sampling, this overprecision is remedied as deep nodes collapse (especially under

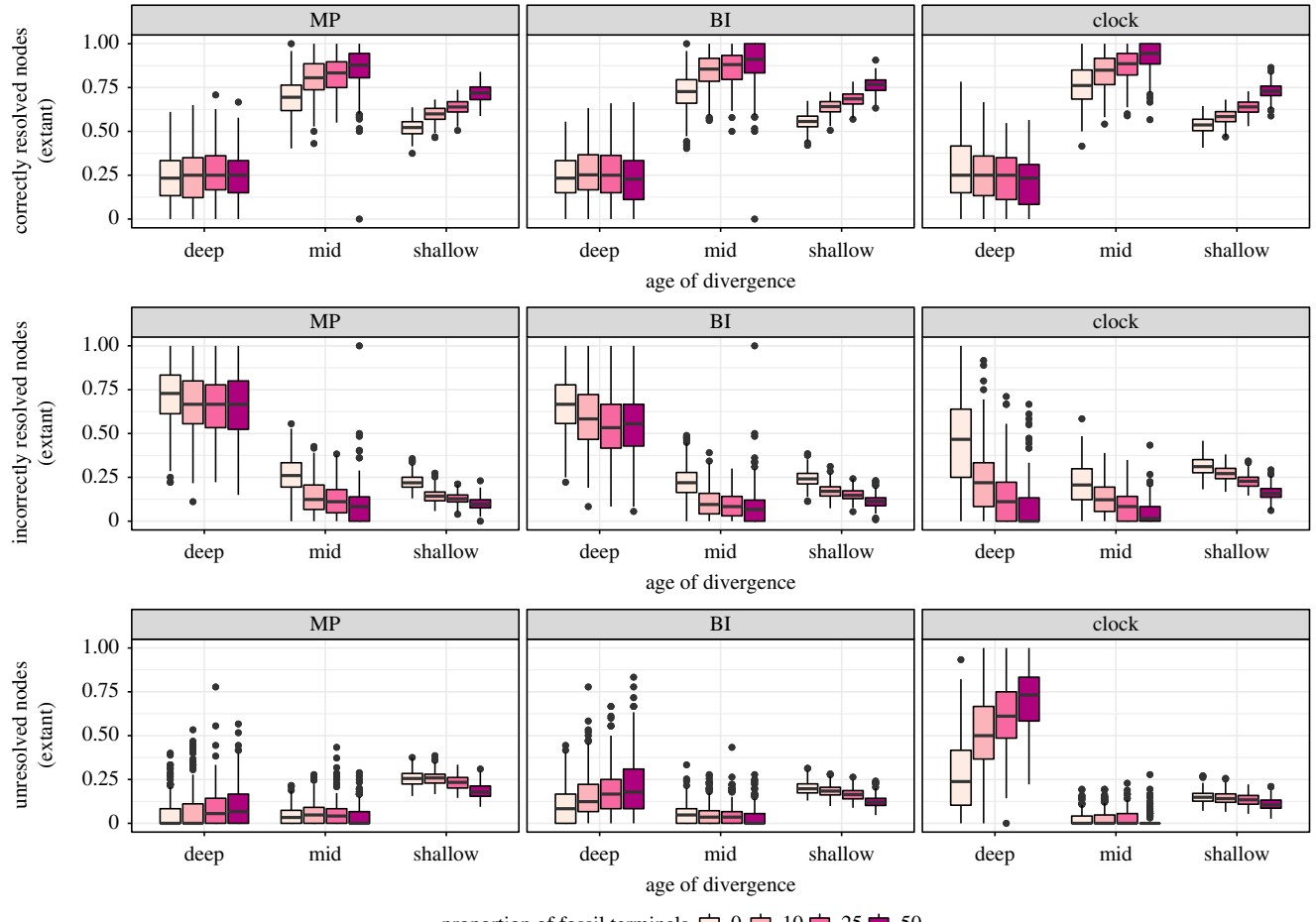

**Figure 3.** Effect of fossils on the way relationships among extant taxa are resolved (from top to bottom: correctly, incorrectly and unresolved). Results correspond to the high missing data condition, as this was designed to mimic empirically realistic datasets. Nodes connecting extant taxa were grouped into three time-bins of equal duration, representing deep-, mid- and shallow divergences. (Online version in colour.)

tip-dated inference), increasing the accuracy of the resulting topologies. Therefore, fossils help recognize the high uncertainty associated with resolving such complicated phylogenetic questions with the use of small morphological datasets. Although missing data decreases both accuracy and precision, it has little effect on the resolution (or lack thereof) of deep nodes, and thus impacts quartet-based measures minimally (figure 1).

In recent years, probabilistic approaches that explicitly model the processes of morphological evolution and species diversification have increased in prominence. Our results corroborate recent studies [20–24] by suggesting that probabilistic methods recover consensus topologies more similar to true trees compared to MP (figure 2). This pattern holds true across all conditions explored, but becomes stronger with increasing levels of missing data, which adversely impact parsimony more than probabilistic methods (electronic supplementary material, figure S8). Even though Bayesian approaches have been criticised for their handling of incompletely coded morphological characters [52], we find missing data have a comparatively small effect on Bayesian consensus trees. By contrast, realistic levels of missing data in MP analyses completely negate the benefit of a more thorough fossil sampling.

Probabilistic methods of inference can also directly employ the morphological and stratigraphic information from fossils to inform divergence times, allowing for greater flexibility in the integration of molecular and palaeontological data [53]. Such integrative, total-evidence, approaches have provided unique insights into the origin and evolution of numerous lineages [54–57]. However, the ways in which morphological and stratigraphic information interact to determine tree topology is an active research area that is arguably in its infancy [17,27,58]. For example, while some improvements in tree topology have been found when fossil ages are included in the process of phylogenetic inference [27], temporal data can also override morphological signals in potentially detrimental ways [17,59,60]. Here, we show that Bayesian tip-dated methods which make use of stratigraphic information significantly outperform undated methods across most of the conditions we explored, indicating that stratigraphic ages provide important phylogenetic information [13,17,61,62]. However, the relative benefit of tip-dating seems to diminish in the presence of realistic levels of missing data (electronic supplementary material, figure S8), to the point that tip-dated topologies of entirely extinct clades are not significantly better than undated ones (figure 2). Topological changes induced by tip-dating fossil clades (e.g. [63,64]) should therefore be considered cautiously. Furthermore, tip-dated inference under the FBD struggles to infer the position of fossil terminals (as also shown by [56]), and is the least accurate method for placing young fossils (electronic supplementary material, figure S9).

Fossils often overturn relationships among extant taxa, even when highly incomplete [10]. The way they do so, however, depends on the relative age of extant clades: fossils increase accuracy of mid- to shallow nodes, while decreasing overprecision among deep nodes, including those involved in

ancient radiations (figure 3). Several authors have hypothesized such radiations are cases where a strong contribution from fossils might be expected, as they represent the only taxa that can directly sample the radiation event, and have character states that are less burdened by subsequent evolutionary change [31,65]. Our results suggest instead that such ancient events of rapid diversification may be out of the reach of morphological signal, and that favoured resolutions are likely to stem from convergences acquired later in evolutionary history (especially if fossils are not sampled). However, even though fossils do not help resolve ancient nodes in phylogenies, they nonetheless play an important role by inducing the collapse of the nodes involved in these deep divergences, mitigating the misleading signal provided by extant taxa. This effect is stronger when their temporal information is incorporated, and is a major driver of the improved accuracy of tip-dated phylogenies.

## 5. Conclusion

Reconstructing evolutionary processes that occurred in the distant past benefits from integrating molecular and palaeontological data [3,66,67], a goal that is facilitated by total-evidence dated inference [14,53–58,60]. Within this framework, extracting accurate phylogenetic signal from morphological datasets is crucial, as morphology has been shown to impact tree topology [68] and divergence-time estimates [60,69] even when combined with genome-scale datasets. Our analyses show that this goal can be achieved through increased fossil sampling, as both the morphological and stratigraphic information from fossils positively impact tree topology.

However, tip-dating is also sensitive to the presence of missing data [58], and thus combining fossils with more complete morphological and/or molecular data from living relatives (if these exist), is likely to result in the most accurate inference of phylogenetic relationships.

Data accessibility. Electronic supplementary material including Detailed Methods, figures and tables. TREvoSim v. 2.0.0 code and binaries are available from [33] and source code is archived at [32]. Empirical and simulated datasets, inferred trees, R code and simulation settings are available from the Dryad Digital Repository: https://doi.org/10.5061/dryad.4xgxd2585 [34].

Authors' contributions. N.M.K., L.A.P. and R.J.G. conceived and designed the study and wrote the manuscript. R.J.G. performed the simulations and coded TREvoSim. N.M.K. and R.J.G. performed the phylogenetic analyses. N.M.K. wrote the analytical pipeline in R, with minor contributions from R.J.G. N.M.K. performed the downstream statistical analyses. All authors gave final approval for publication and agree to be held accountable for the work performed therein.

Competing interests. We declare we have no competing interests.

Funding. This work was supported by the Natural Environment Research Council (grant no. NE/T000813/1 awarded to R.J.G.). N.M.K. was supported by a Yale University fellowship. L.A.P. was supported by an early career fellowship from St Edmund Hall, University of Oxford.

Acknowledgements. We thank Derek E. G. Briggs for helpful comments that greatly improved the clarity of this manuscript, Martin R. Smith for help with the interpretation of quartet distances, Joe Keating for discussion regarding analyses, and Serjoscha Evers for providing a preliminary empirical dataset for turtles. We are also grateful for the comments of two anonymous reviewers. The HTCondor service was provided by the IT Services Research Infrastructure team at the University of Manchester, and runs the HTCondor software developed by the CHTC Team at UW-Madison, Wisconsin. We acknowledge the Willi Hennig Society for providing free access to TNT.

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
