## [Peer Review File · Proceedings of the Royal Society B: Biological Sciences]

Review History

RSPB-2021-0044.R0 (Original submission)

Review form: Reviewer 1

Recommendation

Major revision is needed (please make suggestions in comments)

Scientific importance: Is the manuscript an original and important contribution to its field?

Good

General interest: Is the paper of sufficient general interest?

Good

Quality of the paper: Is the overall quality of the paper suitable?

Marginal

Is the length of the paper justified?

Yes

Should the paper be seen by a specialist statistical reviewer?

No

Do you have any concerns about statistical analyses in this paper? If so, please specify them explicitly in your report.

No

It is a condition of publication that authors make their supporting data, code and materials available - either as supplementary material or hosted in an external repository. Please rate, if applicable, the supporting data on the following criteria.

Is it accessible?

No

Is it clear?

N/A

Is it adequate?

N/A

Do you have any ethical concerns with this paper?

No

Comments to the Author

This manuscript uses a simulated evolutionary process and a few common methods of phylogenetic inference to assess the impact of fossil data on the accuracy and resolution of inferred phylogenies. This study thus adds to previous work, based on empirical analysis and simulation, showing the positive role of fossils in phylogenetic inference. It would be nice to have more of an intuitive explanation why the authors get the results they see. The Discussion states that simulated fossil data improve accuracy of shallow and intermediate nodes rather than deep ones, but in a way that explanation just kicks the can down the road. Why are different node depths affected differently? Answering that could increase the impact of this work.

It has long been understood that alternative methods for phylogenetic inference make assumptions, often hidden, about the nature of evolutionary processes. The Keating et al. paper (ref. 24 in the present MS) provided a simple 2 x 2 combination of simulation methods (MBL vs. TREvoSim) and inferential methods (Parsimony vs. Bayesian). They found that both the simulation method and the inferential method affected accuracy of inferred phylogenies. Bayesian trees are overall more accurate for either simulation method. And MBL evolution yields trees that are more accurately recovered by either inferential method (See especially pp. 903-904 of ref. 24. Other systematic differences in data properties between the two simulation methods are also discussed.) Keating et al. conclude (p. 910): "...rather than focusing on the relative merits of Bayesian or parsimony analyses of morphological data, future analyses might be better directed at identifying modes and patterns of morphological evolution that can ultimately be incorporated into more nuanced models for phylogenetic inference."

Given the results of Keating et al., I have trouble understanding two aspects of the present MS.

- First, the authors seem to imply that TREvoSim, as an organism-based simulation method, should produce results that are not biased in favor of any particular method of inference. See the first three sentences on p. 6 of the MS. In particular, the conclusion that TREvoSim "...thus should be unbiased towards alternative inference approaches" seems contradicted by the Keating et al. work.

- Second, given prior evidence, including ref. 24 and papers cited therein, that simulation method matters, why confine this study to just one simulation procedure?

Review form: Reviewer 2

Recommendation

Accept with minor revision (please list in comments)

Scientific importance: Is the manuscript an original and important contribution to its field?

Excellent

General interest: Is the paper of sufficient general interest?

Excellent

Quality of the paper: Is the overall quality of the paper suitable?

Excellent

Is the length of the paper justified?

Yes

Should the paper be seen by a specialist statistical reviewer?

No

Do you have any concerns about statistical analyses in this paper? If so, please specify them explicitly in your report.

No

It is a condition of publication that authors make their supporting data, code and materials available - either as supplementary material or hosted in an external repository. Please rate, if applicable, the supporting data on the following criteria.

Is it accessible?

Yes

Is it clear?

Yes

Is it adequate?

Yes

Do you have any ethical concerns with this paper?

No

Comments to the Author

This is an exceptionally interesting manuscript that details a series of simulations testing whether the inclusion of (1) fossil tips and (2) fossil dates can improve the accuracy of morphological phylogenies. Indeed, both fossil tips and tip dates improve the accuracy, although unfortunately not the precision in resolving deep rapid radiating nodes. The results also reinforce that the impact of missing data is limited (in BI approaches, especially undated). I read the preprint eagerly when it was released, and the results are easy to understand in this version as well. However, as a reviewer trying to dig into the details of the methods, I recommend minor edits to explain the tree simulation method.

Simulated matrices were created using a new/recently introduced method called TrevoSim, which (based on my reading of the Keating paper) relies on a process based model of morphological evolution, rather than character optimization (as in both MP and BI). What is being simulated seems to be the base of a birth-death model with natural selection (altering the

probability of speciation versus extinction). This method allows simulation of a rapid radiation, which I think would have been impossible to achieve using both a character matrix and a traditional phylogenetic reconstruction method.

I think the analyses are very strong, but am having some trouble really understanding the tree simulation method as it is new and the text really assumes the reader is strongly familiar with the previous paper. I am not even sure my understanding is accurate. In addition to text edits, it would be helpful to have a simple diagram explaining what was done in TrevoSim and what was then done in TNT/MrBayes. I believe it is that the character matrices were in TrevoSim and then the phylogenies themselves reconstructed traditionally, but the first sentence of the methods in the main paper contradicts that guess.

The above sounds like a really negative criticism, but it is not, I do think the results are clear, easy to understand, and exciting in terms of impact of fossils and missing data (and actually understanding these does not truly require understanding the exact method of simulation).

It is nice to read a paleontology manuscript that mentions where the data are uploaded and how the code can be accessed. Interestingly the supplement mentions some files being in the electronic supplementary material rather than Dryad, so clarifying that would be helpful.

Decision letter (RSPB-2021-0044.R0)

25-Feb-2021

Dear Mr Mongiardino Koch:

Your manuscript has now been peer reviewed and the reviews have been assessed by an Associate Editor. The reviewers' comments (not including confidential comments to the Editor) and the comments from the Associate Editor are included at the end of this email for your reference. As you will see, the reviewers and the Editors have raised some concerns with your manuscript and we would like to invite you to revise your manuscript to address them.

Reviewer 1 in particular notes some inconsistencies in interpretation between the present MS and Keating et al 2020, which the present study is largely dependent on. Given that there is an author in common between the two studies, we are optimistic that this can be resolved in revision. There are two other points that require substantial revision. First, addressing the other question raised by R1 (Why are different node depths affected differently?), and second, as raised by R2, a better description of the workflow and simulation method.

Please also note that 1 reviewer commented on difficulties running your simulations as provided. Furthermore, sharing via a private Google Drive link is not sustainable in the long-term and a

better solution such as provision of software on Dryad fits our Open Science policies-- please amend this situation in revisions.

Research ethics:

Use of animals and field studies:

It is a condition of publication that you make available the data and research materials supporting the results in the article. Please see our Data Sharing Policies (<https://royalsociety.org/journals/authors/author-guidelines/#data>). Datasets should be deposited in an appropriate publicly available repository and details of the associated accession number, link or DOI to the datasets must be included in the Data Accessibility section of the article (<https://royalsociety.org/journals/ethics-policies/data-sharing-mining/>). Reference(s) to datasets should also be included in the reference list of the article with DOIs (where available).

All supplementary materials accompanying an accepted article will be treated as in their final form. They will be published alongside the paper on the journal website and posted on the online figshare repository. Files on figshare will be made available approximately one week before the

accompanying article so that the supplementary material can be attributed a unique DOI. Please try to submit all supplementary material as a single file.

Please submit a copy of your revised paper within three weeks. If we do not hear from you within this time your manuscript will be rejected. If you are unable to meet this deadline please let us know as soon as possible, as we may be able to grant a short extension.

Best wishes,
Dr John Hutchinson, Editor
mailto:proceedingsb@royalsociety.org

Reviewer(s)' Comments to Author:

Referee: 1

Comments to the Author(s)

This manuscript uses a simulated evolutionary process and a few common methods of phylogenetic inference to assess the impact of fossil data on the accuracy and resolution of inferred phylogenies. This study thus adds to previous work, based on empirical analysis and simulation, showing the positive role of fossils in phylogenetic inference. It would be nice to have more of an intuitive explanation why the authors get the results they see. The Discussion states that simulated fossil data improve accuracy of shallow and intermediate nodes rather than deep ones, but in a way that explanation just kicks the can down the road. Why are different node depths affected differently? Answering that could increase the impact of this work.

It has long been understood that alternative methods for phylogenetic inference make assumptions, often hidden, about the nature of evolutionary processes. The Keating et al. paper (ref. 24 in the present MS) provided a simple 2 x 2 combination of simulation methods (MBL vs. TREvoSim) and inferential methods (Parsimony vs. Bayesian). They found that both the simulation method and the inferential method affected accuracy of inferred phylogenies. Bayesian trees are overall more accurate for either simulation method. And MBL evolution yields trees that are more accurately recovered by either inferential method (See especially pp. 903-904 of ref. 24. Other systematic differences in data properties between the two simulation methods are also discussed.) Keating et al. conclude (p. 910): "...rather than focusing on the relative merits of Bayesian or parsimony analyses of morphological data, future analyses might be better directed at identifying modes and patterns of morphological evolution that can ultimately be incorporated into more nuanced models for phylogenetic inference."

Given the results of Keating et al., I have trouble understanding two aspects of the present MS.

- First, the authors seem to imply that TREvoSim, as an organism-based simulation method, should produce results that are not biased in favor of any particular method of inference. See the first three sentences on p. 6 of the MS. In particular, the conclusion that TREvoSim "...thus should be unbiased towards alternative inference approaches" seems contradicted by the Keating et al. work.
- Second, given prior evidence, including ref. 24 and papers cited therein, that simulation method matters, why confine this study to just one simulation procedure?

Referee: 2

Comments to the Author(s)

This is an exceptionally interesting manuscript that details a series of simulations testing whether the inclusion of (1) fossil tips and (2) fossil dates can improve the accuracy of morphological phylogenies. Indeed, both fossil tips and tip dates improve the accuracy, although unfortunately not the precision in resolving deep rapid radiating nodes. The results also reinforce that the impact of missing data is limited (in BI approaches, especially undated). I read the preprint eagerly when it was released, and the results are easy to understand in this version as well. However, as a reviewer trying to dig into the details of the methods, I recommend minor edits to explain the tree simulation method.

Simulated matrices were created using a new/recently introduced method called TrevoSim, which (based on my reading of the Keating paper) relies on a process based model of morphological evolution, rather than character optimization (as in both MP and BI). What is being simulated seems to be the base of a birth-death model with natural selection (altering the probability of speciation versus extinction). This method allows simulation of a rapid radiation, which I think would have been impossible to achieve using both a character matrix and a traditional phylogenetic reconstruction method.

I think the analyses are very strong, but am having some trouble really understanding the tree simulation method as it is new and the text really assumes the reader is strongly familiar with the previous paper. I am not even sure my understanding is accurate. In addition to text edits, it would be helpful to have a simple diagram explaining what was done in TrevoSim and what was then done in TNT/MrBayes. I believe it is that the character matrices were in TrevoSim and then the phylogenies themselves reconstructed traditionally, but the first sentence of the methods in the main paper contradicts that guess.

The above sounds like a really negative criticism, but it is not, I do think the results are clear, easy to understand, and exciting in terms of impact of fossils and missing data (and actually understanding these does not truly require understanding the exact method of simulation).

It is nice to read a paleontology manuscript that mentions where the data are uploaded and how the code can be accessed. Interestingly the supplement mentions some files being in the electronic supplementary material rather than Dryad, so clarifying that would be helpful.

Author's Response to Decision Letter for (RSPB-2021-0044.R0)

See Appendix A.

RSPB-2021-0044.R1 (Revision)

Review form: Reviewer 1

Recommendation

Accept as is

Scientific importance: Is the manuscript an original and important contribution to its field?
Excellent

General interest: Is the paper of sufficient general interest?

Excellent

Quality of the paper: Is the overall quality of the paper suitable?

Excellent

Is the length of the paper justified?

Yes

Should the paper be seen by a specialist statistical reviewer?

No

Do you have any concerns about statistical analyses in this paper? If so, please specify them explicitly in your report.

No

It is a condition of publication that authors make their supporting data, code and materials available - either as supplementary material or hosted in an external repository. Please rate, if applicable, the supporting data on the following criteria.

Is it accessible?

N/A

Is it clear?

N/A

Is it adequate?

N/A

Do you have any ethical concerns with this paper?

No

Comments to the Author

This revision has for all practical purposes addressed my initial concerns. I nonetheless have two suggestions (strictly optional):

1. The main text (p. 5) states that the simulation procedure "...does not rely on any model later employed in the process of phylogeny reconstruction..." I think it would be more accurate to say that it does not *explicitly* rely on a particular phylogenetic-reconstruction model. The simulation is at a micro scale, but that does not make it independent of what's happening at the macro scale (the typical purview of phylogenetic methods). For example, Fig. S1 shows a possible simulation outcome. This clearly suggests a "gradual" rather than "punctuational" model of evolution at the macro scale. I suggest acknowledging the distinction between explicit and implicit reliance on models of phylogenetic inference.

2. Page 4 of the Supplement gives some caveats regarding the simulation of data that conform to empirical examples. I suggest that a very brief summary of these points be incorporated into the main text (probably at p. 5, where the authors state, "We fine tune our simulations...").

It's an interesting paper that will make people think and should provoke further research.

--Michael Foote

Decision letter (RSPB-2021-0044.R1)

12-Apr-2021

Dear Mr Mongiardino Koch

I am pleased to inform you that your manuscript entitled "Fossils improve phylogenetic analyses of morphological characters" has been accepted for publication in Proceedings B. Congratulations!!

A minor suggested ("strictly optional") change is up to you whether to include or not, but please let us know if you do wish to; it seems a good idea.

Data Accessibility section

Open Access

You are invited to opt for Open Access, making your freely available to all as soon as it is ready for publication under a CCBY licence. Our article processing charge for Open Access is £1700. Corresponding authors from member institutions (<http://royalsocietypublishing.org/site/librarians/allmembers.xhtml>) receive a 25% discount to these charges. For more information please visit <http://royalsocietypublishing.org/open-access>.

Your article has been estimated as being 9 pages long. Our Production Office will be able to confirm the exact length at proof stage.

Paper charges

Sincerely,

Dr John Hutchinson
Editor, Proceedings B

Associate Editor:

Board Member: 1

Comments to Author:

It is my view that this paper is sufficiently interesting and novel to warrant publication in PRSB. The revised ms has been reviewed again by one of the original reviewers, who suggested some minor additional caveat; however the major point (about implicit and explicit models) is more an issue of discussion rather than something that necessarily needs to be included with the presentation of the model.

Response to reviews

We thank the referees and editor for their insightful comments. We are grateful for the opportunity to address all of these points, and believe that doing so has allowed us to clarify a number of aspects of points in our manuscript. We outline our response to each comment in red, as well as the changes we have made to the manuscript to reflect these points. You can also find the version of the manuscript with tracked changes at the end of this file.

Comments by Editor

Reviewer 1 in particular notes some inconsistencies in interpretation between the present MS and Keating et al 2020, which the present study is largely dependent on. Given that there is an author in common between the two studies, we are optimistic that this can be resolved in revision. There are two other points that require substantial revision. First, addressing the other question raised by R1 (Why are different node depths affected differently?), and second, as raised by R2, a better description of the workflow and simulation method.

As detailed below in the response to each reviewer's comments, we have addressed all of these points. Although we believe that the discrepancies between the present MS and Keating et al 2020 are partly a misunderstanding, we have nonetheless made clear in this version of the MS why we believe TREvoSim to simulate data that is unbiased towards any methods of inference. We have also incorporated to the conclusions some explanations for why we believe nodes of different ages are impacted differently, and have improved the explanation of the methodological workflow (including simulation, inference and analysis steps) in the supplementary material, including new supplementary figures that provide a visual summary of the procedures.

Please also note that 1 reviewer commented on difficulties running your simulations as provided. Furthermore, sharing via a private Google Drive link is not sustainable in the long-term and a better solution such as provision of software on Dryad fits our Open Science policies-- please amend this situation in revisions.

We firmly agree with this! The Google Drive binaries were only provided as a temporary solution for review to avoid a public release of the software before the paper. As part of our revisions we have completed a public release of TREvoSim 2.0.0: <https://github.com/palaeoware/trevoSim/releases/tag/v2.0.0>. This github release has both source code and binaries for multiple OSs, and our release procedure automatically pushes through to a permanent Zenodo repository: <https://zenodo.org/record/4564524#.YDpmLHX7SV4>, creating a backup, persistent presence for this release. Both of these are linked from the MS. We have also incorporated the link to the online Dryad repository where all files and datasets are deposited.

Comments by Referee 1

This manuscript uses a simulated evolutionary process and a few common methods of phylogenetic inference to assess the impact of fossil data on the accuracy and resolution of inferred phylogenies. This study thus adds to previous work, based on empirical analysis and simulation, showing the positive role of fossils in phylogenetic inference. It would be nice to have more of an intuitive explanation why the authors get the results they see. **The Discussion states that simulated fossil data improve accuracy of shallow and intermediate nodes rather than deep ones, but in a way that explanation just kicks the can down the road. Why are different node depths affected differently? Answering that could increase the impact of this work.**

We're grateful for these comments, and the opportunity to clarify our manuscript as a result. An explanation for this was included in the previous version of the manuscript, i.e. that characters evolving at rates close to those of empirical morphological datasets do not retain sufficient phylogenetic signal to correctly resolve deep and short internodes. This explanation is now emphasized and made more explicit.

It has long been understood that alternative methods for phylogenetic inference make assumptions, often hidden, about the nature of evolutionary processes. The Keating et al. paper (ref. 24 in the present MS) provided a simple 2 x 2 combination of simulation methods (MBL vs. TREvoSim) and inferential methods (Parsimony vs. Bayesian). They found that both the simulation method and the inferential method affected accuracy of inferred phylogenies. Bayesian trees are overall more accurate for either simulation method. And MBL evolution yields trees that are more accurately recovered by either inferential method (See especially pp. 903–904 of ref. 24. Other systematic differences in data properties between the two simulation methods are also discussed.) Keating et al. conclude (p. 910): "...rather than focusing on the relative merits of Bayesian or parsimony analyses of morphological data, future

analyses might be better directed at identifying modes and patterns of morphological evolution that can ultimately be incorporated into more nuanced models for phylogenetic inference."

Given the results of Keating et al., I have trouble understanding two aspects of the present MS.

The points we make here are in keeping with Keating et al. - although these comments highlight that they could have been explained with greater clarity in both this manuscript (which we have improved) and in Keating et al. (which, sadly, it's too late to reword!). The points drawn here from Keating et al. are part of a wider discussion in that paper which highlights **that the simulation procedure matters because of the properties of the simulated data**. Different simulation approaches differ in the nature of the data they produce (e.g. tree shapes, homoplasy measures, etc.), and that impacts on the accuracy/precision of inference methods. In this manuscript, we have made every effort to set the simulation parameters so that they generate data with properties as close to those of empirical datasets as possible.

- First, the authors seem to imply that TREvoSim, as an organism-based simulation method, should produce results that are not biased in favor of any particular method of inference. See the first three sentences on p. 6 of the MS. In particular, the conclusion that TREvoSim "...thus should be unbiased towards alternative inference approaches" seems contradicted by the Keating et al. work.

The nature of data similarity is particularly acute where the model used to generate data from a tree topology is the same as is used to infer it (e.g. using MK to generate data will then favour Bayesian which uses MK to infer the topology from data). This reviewer comment picks up on a section where we highlight that the model used to simulate data is not used by any of the inference methods, and thus there is no expectation that one would be preferred. This does not conflict with Keating et al., as the different simulation models used in that manuscript were shown to differ, but not in a manner that favors different inference methods. We have clarified this in our text (see also below).

- Second, given prior evidence, including ref. 24 and papers cited therein, that simulation method matters, why confine this study to just one simulation procedure?

This also relates to the point that the most important factor is the **properties of the simulated data**. In this contribution we have moved significantly beyond the study of Keating et al. - instead of using multiple models (with differing attributes in simulated data), here we focus on ensuring our data are realistic. Here we tackle the issues highlighted in Keating et al. by demonstrating that the data we produce is empirically realistic. We do so through benchmarking against twelve total evidence analyses. This obviates any need to use multiple models (and then discuss at length the impact these properties have) because we are instead confident that the TREvoSim data herein has the properties expected of phenotypic data, drawn from example clades across the tree of life. As well as removing the need to test multiple models, this provides confidence that our findings are unlikely to reflect the simulation procedure. We have clarified this point in the *Data preparation* section of our supplementary information.

Comments by Referee 2

This is an exceptionally interesting manuscript that details a series of simulations testing whether the inclusion of (1) fossil tips and (2) fossil dates can improve the accuracy of morphological phylogenies. Indeed, both fossil tips and tip dates improve the accuracy, although unfortunately not the precision in resolving deep rapid radiating nodes. The results also reinforce that the impact of missing data is limited (in BI approaches, especially undated). I read the preprint eagerly when it was released, and the results are easy to understand in this version as well. However, as a reviewer trying to dig into the details of the methods, I recommend minor edits to explain the tree simulation method.

Simulated matrices were created using a new/recently introduced method called TrevoSim, which (based on my reading of the Keating paper) relies on a process based model of morphological evolution, rather than character optimization (as in both MP and BI). What is being simulated seems to be the base of a birth-death model with natural selection (altering the probability of speciation versus extinction). This method allows simulation of a rapid radiation, which I think would have been impossible to achieve using both a character matrix and a traditional phylogenetic reconstruction method.

I think the analyses are very strong, but am having some trouble really understanding the tree simulation method as it is new and the text really assumes the reader is strongly familiar with the previous paper. I am not even sure my understanding is accurate. In addition to text edits, it would be helpful to have a simple diagram explaining what was done in TrevoSim and what was then done in TNT/MrBayes. I believe it is that the character matrices were in TrevoSim and then the phylogenies themselves reconstructed traditionally, but the first sentence of the methods in the main paper contradicts that guess.

Thank you for these comments! We have expanded the Data generation section of the Supplementary Information file to better describe the simulation procedure implemented in TREvoSim. We have also added to new supplementary figures which provide a graphical summary of the steps implemented in TREvoSim (Fig. S1) as well as all subsequent steps (sampling, inference, and

analysis; Fig. S5). These figures are referenced in the main text, which we believe will help readers better understand the methods implemented.

The above sounds like a really negative criticism, but it is not, I do think the results are clear, easy to understand, and exciting in terms of impact of fossils and missing data (and actually understanding these does not truly require understanding the exact method of simulation).

It is nice to read a paleontology manuscript that mentions where the data are uploaded and how the code can be accessed. Interestingly the supplement mentions some files being in the electronic supplementary material rather than Dryad, so clarifying that would be helpful.